# Effects of structured small-group student talk as collaborative prewriting discussions on Chinese university EFL students' individual writing: A quasi-experimental study

Hui Helen Li[1,2], Lawrence Jun Zhang [2]*

1 School of Foreign Languages, Wuhan University of Technology, Hongshan District, Wuhan, China,
2 Faculty of Education & Social Work, The University of Auckland, Auckland, New Zealand

* lj.zhang@auckland.ac.nz

**Data Availability Statement:** All relevant data are within the manuscript and its Supporting information files in a Zip folder.

## Abstract

Prior studies have reported inconsistent findings with regard to the effects of small-group student talk on developing individual students' English-as-a-foreign-language (EFL) writing ability. To further explore the question under discussion, we designed a quasi-experimental study that included a pretest, a posttest, and a delayed posttest, and implemented it in two English-major groups at a university in China. We randomly assigned the students to an intervention group and a comparison group to investigate whether employing structured small-group student talk as collaborative prewriting discussions would effectively facilitate individual students' EFL writing development and whether such effects could be retained. The immediate and sustained effects after the quasi-experimental study was completed were measured by the analytic scores on five components of the writing task (content, organization, vocabulary, language, and mechanics) and the holistic writing scores cumulated of all these components. Statistical analyses revealed that the two groups were significantly distinguished by their analytic and holistic scores, indicating that students in the intervention group outperformed their comparison group peers in writing performance. The effects of collaborative prewriting discussions in the form of structured small-group student talk were found statistically significant in facilitating students' writing improvement in the content, organization, vocabulary, and language use, but not mechanics. The effects on content, organization, and vocabulary were retained as seen from the delayed posttest, while those on language use were not. The comparison group showed little improvement in their writing performance across the three tests. We concluded this study with a discussion on the implications for English-as-a-second/foreign-language (L2) writing instruction.

## 1 Introduction

Talk is regarded as "the sea upon which all else floats" [1]. It has also been considered as a key factor for all school learning, including writing, which is arguably the most challenging skill

**Funding:** This study was supported by Hubei Provincial Department of Education of China in the form of a grant awarded to HHL (2018112) and Fundamental Research Funds for the Central Universities of China in the form of a grant awarded to HHL (2021VI007). However, the funders had no role in study design, data collection and analysis, decision to publish, or preparation of the manuscript for publication.

**Competing interests:** The authors have declared that no competing interests exist.

for all learners, both in their first language (L1) and in their second or foreign language (L2). The difficulty exacerbates in the process of learning to write in a foreign language [2–4]. In the writing group, talk creates opportunities for students to help each other with their writing by talking to one another and acting as analytical responders, or critical friends, so that they can be familiarized with using talk as both a source and a means in the process of developing their writing skills [5]. As is evident, talk is generative and supportive of the development and the articulation of ideas for writing prior to the act of transforming the ideas into written text [6]. Although writing is not talk written down, students can be supported in various ways to draw on their talk and develop as writers [7].

As one of the ways for achieving the goal in facilitating students' writing development, small-group student talk is the meaningful discussions among students in small groups who talk about writing tasks prior to their individual writing, which can be categorized into content talk, language talk, organization talk, task-management talk, affective talk, and phatic talk [3]. Unlike "brainstorming" [8], which is a group or individual creativity technique by which attempts are made to gather whichever ideas popping into mind about a specific topic either in written or oral forms without giving or receiving any criticism [9,10], structured small-group student talk is what the teacher structures, which not only allows all participants to express opinions/views on the writing topic, showing agreement or disagreement with one another, but also provides a platform for all participants to generate and evaluate ideas and evidence and select and organize them into a writing plan [11].

Recent studies into L2 writing have seen small-group student talk being increasingly used for developing students' writing ability (e.g., [12–19]). One strand of such research has explored student talk in the form of peer dialogue in pairs or small groups during collaborative writing tasks when students co-authored their written texts. Some of these studies have documented the nature of such dialogue by focusing on either group dynamics in terms of patterns of interaction [14,16–18,20–23] or the language-related episodes (LREs) [24–32]. Others have analysed the effects of student talk on collaborative writing by comparing the co-constructed written texts with those independently produced [11,15,25,31,33–35]. A few studies have particularly probed learners' perspectives on peer dialogue in collaborative writing [12,16,36,37]. Small group student talk has also been successfully used in initial teacher-training rooms in order to find out how these student-teachers learn to teach English (e.g., [38]).

Another strand of research has examined student talk in the form of peer feedback interactions in pairs or small groups after individual or collaborative writing. Some of them have investigated the nature of such interactions concerning what students talk about [39–41] as well as the functions and subjects of peer feedback interactions [42,43]. Some others have focused on how students interact in pair and small group activities for peer feedback and how such interaction affects students' text revisions and influences their writing development [19,23,44–46]. A small number of studies has explored learners' perceptions of peer feedback interactions [47–50].

In summary, in the field of L2 writing studies, we have gained a good understanding of the utility of taking stock of student talk for developing students' collaborative or individual writing skills during writing or after the writing task is completed. Studies targeting L2 pre-writing planning have mainly dwelled on individual planning by examining the effects of task complexity, language proficiency level, or planning conditions on learners' written texts [51–55] instead of investigating the effects of L2 learners' talk on their individual writing performance. Consequently, much less attention, so far, has been paid to small-group student talk before L2 individual writing, particularly when it comes to using structured small-group student talk to plan for L2 individual writing. To advance the current knowledge of such an issue in L2 writing, this study adopted a quasi-experimental design [56] with a pretest, a posttest and a delayed

posttest to address whether structured small-group student talk, used as group discussions for collaborative planning, exerts immediate and sustained effects on Chinese tertiary EFL students' individual writing performance.

## 2 Literature review

### 2.1 Sociocultural theory

Given that group talk is a highly demanding intellectual activity of social interaction, we need to take into consideration how our study can be guided by a particular theory. In this case, we found it apt to resort to Sociocultural Theory [57]. This is because this theory emphasizes that learning and development are situated in social interactions and occur as a learner interacts with other people in the collaborative activities. Under the umbrella of Sociocultural Theory, knowledge is first constructed by interactions of individuals in a social context and is then internalized and used by individuals themselves [58]. To be specific, the construction of such knowledge occurs within the learner's zone of proximal development (ZPD), which refers to the distance between the learner's actual and potential developmental levels [59]. From a sociocultural perspective, social interactions play key roles in learning within ZPD. Through social discussions and collaborations with others, learners have opportunities to process knowledge first externally and then internally. In order for ZPD to work effectively during such processes, scaffolding is needed since it provides types of assistance that can bridge the gap between what learners already know and what they want to know. In addition, collective scaffolding enables learners to get either the expert-novice assistance in unequal situations or the learner-learner assistance in equal situations [60].

Studies on L2 writing framed in Sociocultural Theory have documented the benefits of peer interactions for fostering social interaction, writing development, and knowledge co-construction. Specifically, peer interactions before L2 writing enable learners to collectively scaffold each other, generate ideas, and pool their linguistic resources to solve linguistic problems [3,11,31,61,62]. Peer interactions during L2 writing provide learners with opportunities to promote peer collaboration and practice employing the target language for various functions that can potentially be conducive to language learning [16,24,26,30,33–35,63]. Peer interactions after L2 writing guide students to negotiate meaning, realize the gap between their own actual and potential levels, exchange feedback, facilitate them with advice on what to do for improving their writing, e and support them with strategies for to improving their work [17,45,46,64–66].

Given that verbal interactions are considered as the most effective form of peer interaction for knowledge construction and that discussion is one of the most widely-used forms of verbal interaction [33,67,68], students in the writing group ought to be encouraged to actively talk with one another so that they can have opportunities to scaffold themselves, support their ZPDs, and make explicit the implicit nature of learning to write in a second/foreign language [69]. The features of Sociocultural Theory together with previous studies underpinning such a theory have built a solid rationale for the current study. In line with them, the structured small-group student talk in the current study can provide students with opportunities to scaffold one another and bridge the gaps between their present and expected developmental levels in terms of what they have personally known and what the writing tasks explicitly demanded them to chieve.

### 2.2 Effects of small-group student talk on L2 individual writing

Although many studies have been conducted to investigate the effects of small-group student talk on learning, as reviewed in the sections above, studies into the effects of small-group student talk on L2 individual writing in recent years are insufficient [e.g. 11,15,62,70–76]).

Some of these studies have compared the effects of using or not using small-group student talk and have reported mixed findings. For example, Shin [76] explored the effects by comparing individual and collaborative planning of English learners in a Korean university. It was found that the oral planning group members obtained significantly higher scores than those in the individual planning group on all five analytic measures in the expository writing task, including content, organization, language in use, grammar, and mechanics. Similarly, Neumann and McDonough [11,62], who investigated the relationship between collaborative prewriting discussions and students' English written texts, confirmed that the texts produced following collaborative prewriting discussions were scored higher in terms of content than those produced following individual planning. However, McDonough et al. [15], after analysing students' written English texts under three conditions (collaborative writing, collaborative prewriting, and no collaboration) in a Thai university, pointed out that no significant differences among the three conditions were found regarding the analytic ratings (content, organization, and language). Such findings neither supported Shin's [76] study that proved higher analytic scores with collaborative planning nor accorded with McDonough and De Vleeschauwer's [72] conclusion that individual planning resulted in higher analytic ratings. In another study, McDonough et al. [73] examined whether collaborative prewriting discussions yielded higher accuracy, complexity, or analytic ratings (content, organization, and language) than individual prewriting planning. The results that more accuracy in and higher ratings of the texts written under collaborative planning with discussions than those written during individual planning support the findings reported in Shin [76], Neumann and McDonough [11,62], and McDonough and De Vleeschauwer [72] but discorded with those of McDonough et al. [15].

Other prior studies have examined the effects by comparing using small-group student talk in different modes, such as peer-led versus teacher-led, talk in different languages, face-to-face versus online, among others. Shi [75] investigated whether students' writing was more effective after peer-led discussions, teacher-led discussions, or no discussions prior to their individual writing. Through the analysis of students' written texts and transcriptions of their oral discussions, she found that teacher-led discussions produced shorter texts, while peer-led discussions resulted in longer texts with more verbs indicating status and possession. Although the study highlighted that prewriting discussions positively affected the length of essays and the use of vocabulary in students' writing, no discernible effects on the scores of students' writing were perceived in the three different modes. Pu [74] explored the effects of three prewriting group discussions (Chinese L1 group, English L2 group, Chinese L1 and English L2 group) on the language quality measured by CAF (complexity, accuracy, and fluency) of argumentative compositions produced by first-year Chinese English major students. The study revealed that compared with the other two groups, the English L2 group concentrated more on the task and elicited on-task talk most and off-task talk least, thus produced better written texts with statistically higher mean scores of writing quality. A recent study [71] explored whether there existed different effects of face-to-face and online discussions on students' L2 Chinese individual writing. Although neither modes in Liao's [71] study provided solid evidence for grammar-related meaning negotiations, they did confirm that small-group student talk prompted L2 students to scaffold each other within their ZPDs and pool linguistic as well as ideational resources to better develop their individual writing. Most recently, Jiang et al. [70] compared the effects of face-to-face and computer-mediated prewriting group discussions in a technological university from northern China. By measuring the writing performance, cognitive load ratings, and conversational features of interactions in these two conditions, their study revealed that more negotiations for meaning happened in the face-to-face group which performed significantly better in essay writing and experienced lower levels of the extraneous cognitive load than the participants

in the computer-mediated condition. The findings concerning students' written texts in Pu [74], Liao [71], and Jiang et al. [70] echoed each other but contradicted that of Shi's [75] study which identified no immediate effects on the writing scores of L2 students' written texts.

In summary, certain knowledge has been documented in recent studies confirming that students' group talk created opportunities for them to scaffold within each other's ZPDs so that they could co-construct their knowledge and experience. Despite the fact that small-group student talk were reported to be effective for L2 individual writing, mixed findings were discerned, varying from linguistic measures to analytic ratings. More importantly, relatively less is known about whether such effects could be retained over time, particularly when it comes to the structured small-group student talk which was reported to be helpful for students to elicit more talk about organization and produce more evaluative comments [11]. In order to document a comprehensive and thorough understanding of such effects, additional studies that employ pre-, post-, and delayed post-test measures are needed. By using structured small-group student talk as a pedagogical practice, other than as a linguistically different form from writing, this study aims to bridge the above gaps and answer an overarching question: *Does structured small-group student talk prior to individual writing help Chinese tertiary EFL students produce better texts than those written without it*? In so doing, an intervention was performed in the regularly-scheduled Chinese tertiary EFL writing groups and two sub-questions were addressed:

1. Does structured small-group student talk improve Chinese tertiary EFL students' writing performance in terms of analytic (i.e., content, organization, vocabulary, language use, and mechanics) and holistic scores?

2. Is there any difference in the effect of planning with structured small-group student talk and that without it on students' content, organization, vocabulary, language use, mechanics, and overall scores?

## 3 Materials and methods

A quasi-experimental study [56] was adopted as the research design for the current study. Quasi-experimental studies are frequently used in educational contexts because they are constructed from situations that already exist in the real world (see original idea in [77]). This study specifically aimed at six students in each group for the following two reasons. Firstly, three frequently used types of groups for collaborative interactions have been reported by Strijbos et al. [78], including interactions between pairs (dyads), discussions among three to six members (small groups), and interactions among seven or more members (large groups). The large group sizes of 40 to 100 students per group in Chinese universities make it difficult for the teacher to organize and supervise too many dyads in the group. It is also strenuous for the teacher to make sure each student has chances to talk in large groups within the limited group time. Secondly, since the seating arrangement in most Chinese universities only allows fixed desks and chairs, more groups with fewer students in each group will pose noise problems when all groups are discussing at the same time. Group management is equally, if not more, challenging. Therefore, a decision of small groups with six students per group was made with due consideration of all the challenges mentioned above.

### 3.1 Research context and participants

This study was reviewed and approved by The University of Auckland Ethics Committee on Human Participants. It was conducted in a Chinese university that was approved as stated

above. Written informed consent was obtained from the participants for the publication of any potentially identifiable data included in this article. Details of the context are provided below.

**3.1.1 Research context.** The current study was conducted in a comprehensive university in Central China in a teacher-fronted and test-driven context [79], where a compulsory *English Writing* course was offered to second-year English-major students. Following the ethics requirements for keeping the research site and the participants anonymous and confidential, the university and the participant names are not presented in this paper. This course aimed to improve students' competence in English writing of different genres and train them to take a critical view of what they would be discussing and writing about. The course spanned two semesters in the selected university. Instruction in the first semester mainly focused on paragraph writing and the writing of narrative and expository essays. Instruction in the second semester primarily concentrated on writing argumentative essays for the preparation of national exams for English majors, namely, the Test for English Majors (TEM)–Band 4, which tests students' command of English in various skills, including, reading, writing, listening, speaking, and translation. This study was conducted in the second semester.

**3.1.2 Participants.** All the participants in the current study were recruited using convenience sampling [56]. Altogether, 48 sophomore students majoring in English Language and Literature from two intact *English Writing* groups participated in this study. The voluntary participants all grew up in China with Chinese as their mother tongue. They had studied English previously in primary and secondary schools for an average of 10.6 years (SD = 1.2). Having finished the same university coursework, the participants were admitted by the English Department of the selected university as intermediate-level language learners before their enrolment in this *English Writing* course in the second semester of their second year. The course instructor was an associate professor of English language with a Ph.D. degree in applied linguistics. She had been teaching English-major students for 8 years with 5 years of experience in teaching English writing.

The two parallel groups were randomly assigned to an intervention group (n = 24) and a comparison group (n = 24). There were 19 female participants and 5 male participants in the intervention group between the ages of 18 and 20 (M = 19.3, SD = .76). Participants in the comparison group included 16 females and 8 males between the ages of 18 and 21 (M = 19.5, SD = .93). Since teacher-fronted talk is dominant in Chinese tertiary EFL writing groups [80–82], none of these participants had any previous experience of using small-group student talk as collaborative prewriting discussions in any writing-related groups.

## 3.2 Research instrument

**3.2.1 Writing tests.** As the most commonly tested writing genre in both national and international language tests for Chinese university students [83] and a widely-acknowledged assessment for L2 learners' writing proficiency [84,85], argumentative writing was selected as the genre for the writing tests. We decided on using the argumentative genre as the writing task also because it has been regarded as the most important genre during university study for English-major students.

All the participants took part in the pretest, the posttest and the delayed posttest with the same writing tasks as used before, at the end of four weeks after the intervention (see Table 1). The writing task for these tests came from the database of China's National English as a Foreign Language Test—Test for English Majors—Band 4 (TEM-4), which has been reported to have high validity and reliability [83,86]. TEM-4 is a nationally standardized annual test that is taken by Chinese university English-major students at the second semester of their second

year. Such a test is claimed to be drawn on students' daily life and is generally believed to be familiar and fair to each student [85]. The selection of the argumentative writing task from the TEM-4 database offered participants opportunities to prepare for the test, which could not only arouse their interest but also boost their enthusiasm to engage in this study.

All the three tests were administrated by the first author of the current study. During each test, no external resources were allowed for the purpose of obtaining data on participants' real English writing performance. Apart from that, the writing prompt, test time, and procedures were kept constant in both groups regarding the pretest, the posttest and the delayed posttest.

## 3.3 Procedures of Instrumentation

**3.3.1 Intervention for data collection.** Both groups met the course instructor in a regular classroom setting. Each group had two 45-minute sessions per week with a total of 32 sessions in a 16-week semester. Both groups followed the same teaching syllabus and plan as required by the English Department of the selected university. A theme-based textbook named *Writing Critically III—Argumentative Writing* was used as the course book, which centred on training students to write argumentative essays. The textbook is specifically designed for English-major undergraduates with orientations in English Language and Literature, Translation, and Business English. It has been included in the list of approved textbooks for use in China's National Standard Textbooks for English Major Students in Tertiary Institutions.

Given that structured prewriting tasks were proven to be more effective in engaging L2 writers in critical evaluation of their ideas and organization than the naturally-occurring peer-led discussions [11,73], the writing topic for the tests was accompanied by an extra section (see S1 Appendix). This structured section was adapted from Neumann and McDonough [11] and it consisted of three parts: (a) giving opinions on agreeing or disagreeing with the writing topic, (b) generating and evaluating ideas and evidence that are for and against the writing topic, and (c) selecting and organizing ideas and evidence into a writing plan.

A practice session was conducted to familiarize participants with the processes. Altogether, five rounds of structured small-group student talk were administered as the intervention and were recorded by the instructor with digital recording devices (see Table 1). The writing topic for the practice session came from the theme-based textbook mentioned above. The other five writing topics for intervention were randomly selected from the past TEM-4 test battery. Before the first practice session began, participants in the intervention group were asked to form into four groups of six students based on the method of self-selection. This method was reported to be effective for facilitating participants in group collaboration with a higher sense of goal commitment and group accomplishment [87]. Participants were told that they would stay in their assigned group throughout all the six rounds.

During each round of intervention, while participants in the intervention group were engaged in structured small-group student talk for planning, participants in the comparison

**Table 1. Procedures of the study.**

| Week | Intervention Group | Comparison Group | Writing Topic |
|------|--------------------|------------------|---------------|
| 1 | Pretest (40 min) | Pretest (40 min) | Test task (see S1 Appendix) |
| 2 | Practice session of small-group student talk for planning (20 min) + individual writing (40 min) | Individual planning (20 min) + individual writing (40 min) | Practice task from the course textbook |
| 3, 5, 7, 9, 11 | Intervention sessions | Planning individually as usual | Intervention tasks from Chinese TEM-4 battery |
| 12 | Posttest | Posttest | Test task (see S1 Appendix) |
| 16 | Delayed posttest | Delayed posttest | Test task (see S1 Appendix) |

group were performing individual planning as usual. Specifically, in each session, each small group in the intervention group first talked for 20 minutes based on the structured writing task and then separated to write the task individually for 40 minutes. However, participants in the comparison group planned individually for 20 minutes following the same structured writing task as used in the intervention group and after that they proceeded to individual writing of the task for 40 minutes. During these sessions, neither the intervention group nor the comparison group was allowed to use any external resources. Meanwhile, the course instructor mainly remained silent as an observer unless students particularly asked for her help.

The decision to administer 20 minutes for the structured small-group student talk and 40 minutes for individual writing was based on Shi's [75] research, which reported that students could generate sufficient talk in group discussions in 20 minutes and write drafts of essays of a reasonable length within 40 minutes. Recently, researchers such as McDonough and De Vleeschauwer [73] and Neumann and McDonough [11] also arranged a similar timeframe in their studies when group talk was used as collaborative prewriting discussions for individual writing. Meanwhile, considering that TEM-4 required students to finish the writing section with no fewer than 200 words in 40 minutes, the course teacher preferred to follow the time arrangement of the standard TEM-4 test. Therefore, the current study allotted 40 minutes for individual writing in both groups.

In total, 288 students' individual writing samples (48×2×3) written by participants in the intervention group and comparison group were collected from the pretest, posttest and delayed posttest. These written texts were used to determine the effects of small-group student talk on students' argumentative writing performance in terms of analytic (content, organization, vocabulary, language use, and mechanics) and holistic scores.

## 3.4 Data analysis

**3.4.1 Interrater reliability.** Although TEM-4 rubric is nationally acknowledged in China as an official tool to assess English-major sophomore students' language proficiency, the current study intended to evaluate the writing competence of English learners in China with a more internationally recognized rubric. In so doing, the study might be able to gain a more complete understanding of their writing ability. Also, it might help extend the research to reach a broader and global audience, who can bench-mark Chinese students' writing performance in relation to the widely used measures in many other contexts. In this regard, the well-established and widely used writing rubric that was originally developed by Jacobs, Zinkgraf, Wormuth, Hartfiel, & Hughey [88] and modified and updated by Hedgcock and Lefkowitz [89] was used to rate and determine the overall quality of students' written texts, both holistically and analytically. The rubric comprises five component areas on a 100-point scale, including content (13–30), organization (7–20), vocabulary (7–20), language use (5–25), and mechanics (2–5). Each component of the scale consists of the following four bands: excellent to very good, good to average, fair to poor, and very poor (see S2 Appendix).

Two Chinese raters, who held their Ph.D. degrees in second language acquisition or applied linguistics from well-known universities overseas and had no direct involvement in any other aspects of the current study, rated all the written texts. A blind assessment was implemented in which the raters did not know which group of students they were rating, nor did they know if they were rating a pre-, post-, or a delayed post-test written text. A subset of the total texts (48/288 of the texts or about 17%) was randomly selected and rated by the raters to check for rating consistency and reliability. The final score of each text was the aggregated average value of the ratings given by the two raters. The interrater reliability was calculated and finally established at .955 for holistic scores, which could be considered acceptable. It has been well acknowledged

that if inter-rater reliability reaches .70, it is considered as acceptable [90]. As for the analytic scores in terms of the five components, the interrater reliability for each component was also satisfactory (content, r = .927; organization, r = .878; vocabulary, r = .885; language use, r = .878; mechanics, r = .789).

**3.4.2 Statistical analyses.** To address the two research questions, SPSS 23.0 was employed for statistical analyses. Three aspects including normality, missing values and outliers were examined prior to statistical analysis. No missing values and outliers were found. All the data of this study were normally distributed because the z-scores of skewness and kurtosis did not exceed 1.96 [91] when a normality check was performed. Then independent-samples *t*-tests were applied to investigate whether there were any effects of small-group student talk on students' individual writing in terms of the overall quality and the quality of argument between the intervention group and the comparison group. After that, one-way repeated measures ANOVAs were conducted to examine within-subjects differences in each group. A series of paired samples *t*-tests were employed when significant changes were found from the one-way ANOVAs with repeated measures to explore whether the intervention group displayed significantly better results than the comparison group. During the multiple comparisons, a Bonferroni correction was used to avoid Type I errors. As for the interpretation of the effect sizes, this study followed Cohen's [92] criteria that *d* values of .20, .50, and .80 and partial $\eta 2$ values of .01, .06, and .14 were deemed as small, medium, and large, respectively.

## 4 Results

### 4.1 Effects on overall writing quality

To address the two research questions, descriptive statistics of students' holistic and analytic scores between the intervention group and the comparison group across the three tests were calculated (see Table 2). Independent samples *t*-tests were run in order to check the baseline conditions of the two groups at the beginning of the intervention.

The results (see Table 3) showed that no significant between-subject differences were found in different measures at the time of pretest (overall, *p* = .943; content, *p* = .858; organization, *p* = .604; vocabulary, *p* = .528; language use, *p* = .882; mechanics, *p* = .526). However, statistically significant differences were found between the intervention group and the

**Table 2. Descriptive statistics for holistic and analytic scores of overall writing quality across tests.**

| Measures | Group | Pretest | | Posttest | | Delayed posttest | |
|---|---|---|---|---|---|---|---|
| | | Mean | SD | Mean | SD | Mean | SD |
| Overall | CG | 72.92 | 3.73 | 75.63 | 3.68 | 73.75 | 3.58 |
| | IG | 73.00 | 4.26 | 78.29 | 3.98 | 76.58 | 4.39 |
| Content | CG | 21.88 | 1.26 | 22.46 | 1.59 | 22.25 | 2.13 |
| | IG | 21.79 | 1.89 | 23.75 | 2.05 | 22.92 | 2.30 |
| Organization | CG | 16.25 | 1.03 | 16.88 | 1.45 | 16.38 | 1.01 |
| | IG | 16.42 | 1.18 | 17.79 | 1.41 | 17.38 | 1.01 |
| Vocabulary | CG | 15.58 | 1.32 | 15.96 | 0.96 | 15.88 | 1.26 |
| | IG | 15.83 | 1.40 | 16.96 | 1.16 | 16.50 | 1.32 |
| Language Use | CG | 16.08 | 0.88 | 16.58 | 1.06 | 16.13 | 0.85 |
| | IG | 16.00 | 1.02 | 16.67 | 1.47 | 16.33 | 1.20 |
| Mechanics | CG | 3.13 | 0.95 | 3.50 | 1.02 | 3.46 | 0.88 |
| | IG | 2.96 | 0.86 | 3.13 | 1.23 | 3.21 | 0.93 |

*Note.* CG = comparison group; IG = intervention group; SD = standard deviation.

**Table 3. Between-subjects comparisons of holistic and analytic scores of overall writing quality across tests.**

| Measures | Pretest | | Posttest | | Delayed posttest | |
|---|---|---|---|---|---|---|
| | *t* | *p* | *t* | *p* | *t* | *p* |
| Overall | -.072 | .943 | -2.409 | **.020**\* | -2.450 | **.018**\* |
| Content | .180 | .858 | -2.442 | **.019**\* | -.2.060 | **.045**\* |
| Organization | -.552 | .604 | -2.214 | **.032**\* | -3.418 | **.001**\* |
| Vocabulary | -.636 | .528 | -2.222 | **.031**\* | -1.678 | .100 |
| Language use | .303 | .763 | -.226 | .822 | -.693 | .492 |
| Mechanics | .639 | .526 | 1.151 | .256 | .954 | .345 |

\*p<.05.

comparison group in the scores of overall writing quality in the immediate posttest ($t$ = -2.409, $p$ = .020, $d$ = -.70) and the delayed posttest ($t$ = -2.450, $p$ = .018, $d$ = -.71). In other words, although students in the intervention group did not achieve significantly better performance than the students in the comparison group on the pretest, they did perform significantly better on the posttest immediately after the intervention and on the delayed posttest four weeks after the intervention.

To determine whether the overall writing quality differed significantly within each group at the time of the immediate posttest and the delayed posttest, one-way repeated measures ANOVAs were used. Results showed that the scores of overall writing quality changed significantly over time in both the intervention group ($F(2, 46)$ = 55.616, $p$<.001, partial $\eta^2$ = .707) and the comparison group ($F(2, 46)$ = 8.507, $p$ = .001, partial $\eta^2$ = .270).

To further explore the within-subjects differences in each group, paired samples t-tests were employed, and Bonferroni correction was applied ($p$ = .017). The results indicated that the scores of the overall writing quality of the intervention group improved with large effect sizes from the pretest to the immediate posttest ($p$<.001, $d$ = -1.80) and from the pretest to the delayed posttest ($p$<.001, $d$ = -1.80). There was also a medium size effect from the posttest to the delayed posttest ($p$ = .003, $d$ = 0.68). In contrast, within the comparison group, significant improvement only showed up from the pretest to the immediate posttest ($p$ = .001, $d$ = -0.81), but it did not manifest neither from the immediate posttest to the delayed posttest ($p$ = .022) nor from the pretest to the delayed posttest ($p$ = .128). Such results revealed that structured small-group student talk effectively promoted students in the intervention group to gain significantly higher scores across the three tests concerning the overall writing quality.

## 4.2 Effects on content

Between-subjects comparisons using independent samples *t*-tests exhibited significant differences regarding content scores between the intervention and comparison groups in the immediate posttest ($t$ = -2.442, $p$ = .019, $d$ = -0.70) and the delayed posttest ($t$ = -2.060, $p$ = .045, $d$ = -0.06) (see Table 3). Considering that both groups performed similarly in content at the beginning of the intervention ($t$ = .180, $p$ = .858), such results proved that students in the intervention group benefitted more from the structured small-group student talk in this measure.

According to one-way repeated measures ANOVAs, content scores changed significantly over time in the intervention group ($F(2, 46)$ = 18.356, $p$<.001, partial $\eta^2$ = .444). However, it was not a case in the comparison group ($F(2, 46)$ = 2.129, $p$ = .131, partial $\eta^2$ = .085). A further examination of within-subjects differences using paired samples *t*-tests found that small-group student talk helped students in the intervention group facilitate content from the pretest to the

immediate posttest ($p$<.001, $d$ = -1.29). In addition, the beneficial effect of structured small-group student talk on students' individual writing regarding content could be retained in the delayed posttest (p<.001, $d$ = -0.90).

## 4.3 Effects on organization

As shown in Table 3, the two groups had similar performance in organization at the outset of the study ($t$ = -.522, $p$ = .604). However, the intervention group outperformed the comparison group in this measure in both the immediate posttest ($p$ = .032), with a medium effect size ($d$ = -0.64), and the delayed posttest ($p$ = .001), with a large effect size ($d$ = -0.99). The positively significant differences indicated that the intervention group gained higher organization scores in their individual writing because of their engagement with the structured small-group student talk.

One-way repeated measures ANOVAs were also run. Results showed that there were statistically significant changes across tests in terms of organization scores both in the intervention group ($F$ (2, 46) = 8.832, $p$ = .001, partial $\eta^2$ = .277) and the comparison group ($F$ (2, 46) = 3.409, $p$ = .042, partial $\eta^2$ = .129). A series of paired samples $t$-tests with Bonferroni correction ($p$ = .017) revealed that the intervention group improved significantly in organization with a close-to-large size effect from the pretest to the immediate posttest ($p$ = .001, $d$ = -0.79). Besides, such an effect was maintained from the pretest to the delayed posttest ($p$ = .006, $d$ = -0.62). However, the organization scores for the comparison group did not demonstrate significant improvement in their writing scores across the three tests (pre- vs. immediate posttest, $p$ = .040; immediate post- vs. delayed posttest, $p$ = .069; pre- vs. delayed posttest, $p$ = .560). These significant comparisons suggest that structured small-group student talk had significant effects on organization in students' individual writing.

## 4.4 Effects on vocabulary

According to independent samples $t$-tests for between-subjects comparisons, no initial differences in vocabulary scores were observed ($t$ = -.636, $p$ = .528) (see Table 3). Although no statistically significant differences were found between the two groups in the delayed posttest ($p$ = .100), the intervention group did show significantly higher vocabulary scores than the comparison group in the immediate posttest ($p$ = .031), with a medium effect size ($d$ = -0.64). This also indicates that the intervention group exhibited better results on this measure than the comparison group.

A further analysis of one-way ANOVA with repeated measures were conducted, which revealed significant changes across tests in both the intervention group ($F$(2, 46) = 8.694, $p$ = .001, partial $\eta^2$ = .274) and the comparison group ($F$(2, 46) = 3.954, $p$ = .026, partial $\eta^2$ = .147). Results of paired samples $t$-tests with Bonferroni correction ($p$ = .017) demonstrated a beneficial effect of structured small-group student talk on vocabulary with a medium-to-large size in the intervention group from the pretest to the immediate posttest ($p$ = .002, $d$ = -0.72). Additionally, this effect was retained with a medium size in the delayed posttest ($p$ = .012, $d$ = -0.55). However, such significant improvement did not appear across the tests in the comparison group.

## 4.5 Effects on language use

The intervention and comparison groups performed similarly with respect to language use in the pretest ($t$ = .303, $p$ = .763). Likewise, these two groups achieved similar performance in this measure in the immediate posttest and the delayed posttest ($t$ = -.226, $p$ = .822; $t$ = -.693, $p$ = .492 respectively) (see Table 3). However, within-subjects comparisons showed that language

use varied significantly over time in both the intervention group ($F$ (2, 46) = 6.133, $p$ = .004, partial $\eta^2$ = .211) and the comparison group ($F$ (2, 46) = 4.479, $p$ = .017, partial $\eta^2$ = .163).

To further investigate such significant differences across tests in each group, paired samples $t$-tests with Bonferroni correction ($p$ = .017) were run. The tests indicated that although the score of language use for the comparison group improved significantly from the pretest to the immediate posttest ($p$ = .015, $d$ = -0.54), it gained a larger effect size in the intervention group ($p$ = .003, $d$ = -0.69). However, such differences did not show up in the intervention and comparison groups neither from the immediate posttest to the delayed posttest ($p$ = .119 and $p$ = .031 respectively) nor from the pretest to the delayed posttest ($p$ = .057 and $p$ = .802 respectively). In other words, such effects were not retained in the delayed posttest.

## 4.6 Effects on mechanics

No statistically significant differences were observed across the three tests between the intervention and comparison groups concerning mechanics in students' individual writing ($t$ = .639, $p$ = .526; $t$ = .1.151, $p$ = .256; $t$ = .954, $p$ = .345 respectively) (see Table 3). The application of one-way repeated-measures ANOVAs revealed that students made no significant progress in mechanics across time, neither in the intervention group ($F$ (2, 46) = .641, $p$ = .531, partial $\eta^2$ = .027), nor in the comparison group ($F$(2, 46) = 2.001, $p$ = .147, partial $\eta^2$ = .080). Such results suggest that structured small-group student talk might have exerted no significant effects on mechanics in students' individual writing.

## 5 Discussion

The effects of structured small-group student talk on students' L2 individual writing performance were measured by the analytic scores of the five components (content, organization, vocabulary, language, and mechanics) as well as the holistic score of these components. The results of statistical analysis indicate that structured small-group student talk had significant facilitating effects on students' L2 individual writing development because students in the intervention group gained higher scores in the posttest. However, such effects varied from one component to another. Specifically, structured small-group student talk was significantly effective in improving students' individual writing performance with respect to content, organization, vocabulary, language use, but not mechanics. Meanwhile, the subscores for content, vocabulary, and organization improved statistically significantly with good effect sizes. In fact, language use was the only component in which students in both groups improved significantly, but the intervention group demonstrated a larger effect size. Mechanics was the only component in which students in neither group made significant progress. Additionally, the effects on content, organization, and vocabulary were retained in the delayed posttest, while the effects on language use were not maintained.

The findings about the immediate effects of structured small-group student talk on L2 individual writing lend support to the results of Shin [76], Pu [74], Neumann and McDonough [11,62], Liao [71], McDonough et al. [73], and Jiang et al. [70] in that texts produced following collaborative prewriting discussions were scored higher than those produced after individual planning. It is inspiring that using small-group student talk as collaborative prewriting discussions to plan for either composition writing [70,71,74,76] or short paragraph writing [11,62,73], can trigger collective scaffolding [93] by virtue of such practice offering L2 learners opportunities to bridge the gaps between each other's actual and potential developmental levels, and as a result, helped improve their individual writing skills.

However, such findings are not consistent with what Shi [75] and McDonough et al. [15] reported. In their studies no discernible effects on the scores of students' individual writing

were found between planning with collaborative prewriting discussions and not providing any opportunities for collaboration. These findings do not endorse McDonough and De Vleeschauwer's [72] claim that individual prewriting resulted in higher analytic ratings regarding the content, organization, and vocabulary. One possible explanation for these differences might lie in different research designs. In Shi's [75] study, she carried out a case study which administered a one-shot treatment with 40 minutes for both the peer-led and teacher-led talk conditions and 60 minutes for the no-talk condition. McDonough et al. [15] and McDonough and De Vleeschauwer [72] also implemented similar research designs with 75 minutes for both the treatment and the comparison groups. These arrangements all varied from this quasi-experimental study which employed pre-, post-, and delayed post-test measures with five rounds of intervention and allocated 60 minutes for both the intervention and comparison groups. Despite that the between-groups comparisons in the case studies can reveal discernible differences between or among different groups, it is difficult to determine if any change within the group itself has taken place as this study did. Such within-group changes are also significant indicators for examining the distinctions between groups. For example, the between-subjects comparisons of the current study found no significant differences between the intervention and comparisons groups across these three tests concerning language use (pretest, $t = .303$, $p = .763$; posttest, $t = -.226$, $p = .822$; delayed posttest, $t = -.693$, $p = .492$). However, the within-subjects comparisons showed that the score of language use for the intervention group improved with a larger size effect from the pretest to the immediate posttest ($p = .003$, $d = -0.69$) compared with that for the comparison group ($p = .015$, $d = -0.54$). Such results suggest a significant distinction between the two groups, which indicated that the differences between the research design of the current study and those of Shi [75], McDonough et al. [15], and McDonough and De Vleeschauwer [72] might have resulted in the inconsistent findings.

Another probable reason could be related to the factor of writing tasks. Specifically, each of the three opinion-essay writing tasks in Shi's [75] study only provided a section of an opinion question together with a brief instruction to inform students to give ideas and suggest one or more solutions for the problem. The paragraph-writing task in the study of McDonough et al. [15] asked students to suggest two practical solutions to the problem, and to support the solutions with logical reasons and relevant supporting details. They also stated that students should have topic sentences and concluding sentences and use appropriate discourse markers. Each writing task in McDonough and De Vleeschauwer's [72] study comprised a section of background information together with a task instruction which informed students to refer to the information in the background section and give suggestions that could address at least two of the problems listed in the background section. Given that structured prewriting tasks were proven to be more effective in engaging L2 writers in critical evaluation of their ideas and organization than the naturally-occurring peer-led discussions [11,73], the current study used a structured writing task, which mainly included a section of a writing topic and an extra part of instructions concerning opinion giving, ideas and evidence generation and evaluation, and writing plan creation. A general examination of the recorded five rounds of small-group student talk revealed that the top three categories were content, language (including vocabulary), and organization, which elucidated the significant improvement in these components in this study.

Finally, such differences might be attributed to the grouping methods and different numbers of group members. Shi [75] randomly selected 4–6 students for each group without considering students' personal relationships, while McDonough et al. [15] and McDonough and De Vleeschauwer [72] both employed a self-select grouping method but with only two students in each group. Unlike each study mentioned above, the current study adopted the self-select grouping method with a larger group size (N = 6). Compared with the random

grouping by the researcher or the classroom teacher, self-select grouping by students themselves could contribute to the ease of communication, facilitation of collaboration, and promotion of satisfaction, which could help students achieve better results [94]. Meanwhile, group sizes can also exert a direct impact on the quality of communication among group members and thus cause different outcomes, because appropriate group sizes may promote high-quality communication, which has been shown to increase group cooperation and pooling of information by individual group members, and thus make group performance exceed that of individual performances [95]. The recorded student talk of this study demonstrated collaborative engagement and good communication among each group when planning for the writing tasks, which might also have contributed to the development of students' L2 individual writing performance.

Concerning the sustained effects of structured small-group student talk on L2 individual writing, the outcomes of the current study resolved the limitations of all the aforementioned studies as to whether repeated opportunities to plan collaboratively using small-group student talk would impact L2 writers' development over time. It is interesting to note that sustained effects were only significantly discerned on content, organization, and vocabulary but not on language use. Such a finding mainly attributes to the teacher-structured writing tasks used in the present study, which specifically required students to express viewpoints clearly, generate ideas for content, and talk about how to select and organize useful ideas. Evidence from students' recorded group talk indicates that during all five rounds of intervention, students frequently produced content talk, asked about unfamiliar or unknown vocabulary, discussed which ideas should be chosen for their writing and how to organize these ideas into their texts. The frequent occurrences of such talk during repeated practice successfully linger in students' mind and help enable them to perform well even when no such practice opportunities are provided. Comparatively, much fewer occurrences of student talk on language use were witnessed during the intervention sessions. Therefore, it is understandable that the effects on language use were not retained. Future research may need to take into consideration a key issue, namely, how to better structure small-group student talk so that the effects of language use could be sustained.

With regard to the immediate and sustained effects on mechanics, the reason why there was a lack of significance for mechanics in both groups could be that the mechanics of writing, which mainly concerns the mastery of writing conventions (e.g., spelling, punctuation, capitalization, and paragraph indentation, etc.), are more straightforward since they represent a relatively limited range of rules and conventions which can be more easily dealt with and mastered by all students in both conditions [37].

## 6 Conclusion

This quasi-experimental study provides empirical evidence uncovering the gap that addresses whether using structured small-group student talk for collaborative planning facilitates Chinese tertiary students' individual EFL writing development. The results revealed immediate and sustained effects of structured small-group student talk on the quality of students' written texts. Specifically, structured small-group student talk significantly improved Chinese EFL learners' individual writing on content, organization, vocabulary, and langue use. Greater effects on content, vocabulary, and organization were found with good effect sizes. Such findings suggest that structured small-group student talk did enable students to collectively scaffold each other for their ZPDs to be triggered and kept as active as possible until they accomplished the writing tasks which were comparatively more difficult when they did them alone [96,97]. However, no discernible effects were observed on mechanics. Besides, although

the study found sustained effects on content, organization, vocabulary, and language use, the effects on language use were not retained.

Understandably, the current study has its own limitations. One limitation is that this study investigated the effects of structured small-group student talk on L2 individual writing performance with a small sampling size (N = 48). Given this limitation, future studies can enlarge the size of sampling to enhance the reliability of the results. Another shortcoming is that this study carried out a repeated writing task for all the three tests, which might have inadvertently helped students better remember more of what they had written in previous tests and to some degree might have influenced student writing performance. For this reason, a similar design with a different argumentative writing task for each test is needed in future research.

Despite the limitations, the current study might have pedagogical implications for teaching argumentative writing in the L2 context. To begin with, given the positive effects of structured small-group student talk on L2 individual writing performance, L2 writing instructors might want to provide enough opportunities for students to engage in structured small-group student talk as a pre-writing discussion for planning their writing [3]. Besides, such a pedagogical practice should be implemented over a longer period with more rounds of sessions than those administered in this study, since enhancing argumentative writing skills needs time and practice [98]. It is important to note that L2 writing instructors' scaffolding and involvement are needed to better structure small-group student talk because structured argumentative writing tasks may help L2 writing instructors exploit the benefits of small-group student talk while maintaining a focus on individual writing development [61]. Therefore, when designing collaborative prewriting discussion activities to prepare students for argumentative writing, L2 writing instructors should take specific teaching goals into consideration and place greater emphasis on the components in which they want their students to improve significantly.

## Supporting information

**S1 Appendix.**
(DOCX)

**S2 Appendix.**
(DOCX)

**S1 Dataset.**
(ZIP)

## Author Contributions

**Conceptualization:** Hui Helen Li, Lawrence Jun Zhang.

**Data curation:** Hui Helen Li.

**Formal analysis:** Hui Helen Li, Lawrence Jun Zhang.

**Funding acquisition:** Hui Helen Li.

**Investigation:** Hui Helen Li, Lawrence Jun Zhang.

**Methodology:** Hui Helen Li, Lawrence Jun Zhang.

**Project administration:** Hui Helen Li.

**Resources:** Hui Helen Li.

**Supervision:** Lawrence Jun Zhang.

**Writing – original draft:** Hui Helen Li.

**Writing – review & editing:** Lawrence Jun Zhang.

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
