## [Decision Letter · Decision Letter 0]

13 Nov 2020

PONE-D-20-31067

Effects of structured small-group student talk on Chinese Tertiary

EFL students’ individual writing: A quasi-experimental study

PLOS ONE

Dear Dr. Zhang,

Thank you for submitting your manuscript to PLOS ONE. After careful consideration, we feel that it has merit but does not fully meet PLOS ONE’s publication criteria as it currently stands. Therefore, we invite you to submit a revised version of the manuscript that addresses the points raised during the review process.

We look forward to receiving your revised manuscript.

Kind regards,

Haoran Xie

Academic Editor

PLOS ONE

Journal Requirements:

2. Thank you for submitting the above manuscript to PLOS ONE. During our internal evaluation of the manuscript, we found significant text overlap between your submission and the following previously published work, of which you are an author.

- https://www.frontiersin.org/articles/10.3389/fpsyg.2020.570565/full

Please revise the manuscript to rephrase the duplicated text, cite your sources, and provide details as to how the current manuscript advances on previous work. Please note that further consideration is dependent on the submission of a manuscript that addresses these concerns about the overlap in text with published work.

3. Please improve statistical reporting and refer to p-values as "p<.001" instead of "p=.000". Our statistical reporting guidelines are available at https://journals.plos.org/plosone/s/submission-guidelines#loc-statistical-reporting.

Reviewers' comments:

Reviewer's Responses to Questions

**Comments to the Author**

1. Is the manuscript technically sound, and do the data support the conclusions?

Reviewer #1: Partly

Reviewer #2: Partly

2. Has the statistical analysis been performed appropriately and rigorously? 

Reviewer #1: Yes

Reviewer #2: Yes

3. Have the authors made all data underlying the findings in their manuscript fully available?

Reviewer #1: Yes

Reviewer #2: Yes

4. Is the manuscript presented in an intelligible fashion and written in standard English?

Reviewer #1: No

Reviewer #2: No

5. Review Comments to the Author

Reviewer #1: This study aims to investigate the effects of pre-writing small-group talk on L2 individual writing, which is interesting and highly relevant. The design of this study is structured and the instruments are reliable. To further improve this paper, the author may consider the points as follows:

1. Is the investigated topic, “small-group student talk before L2 individual writing,” equal to “brainstorming?” If so, the author may consider use the more concise expression; if not, the author may specify the difference between two.

2. The author stated that “much less attention, so far, has been paid to small-group student talk before L2 individual writing, particularly when it comes to using structured small-group student talk to plan for L2 individual writing….” However, to my knowledge, there are many studies focusing on or at least involving the investigation of the effects of pre-writing planning/brainstorming.

3. The content may be preliminary good, but several parts of this article are somehow difficult to read. Redundancy, grammatical inaccuracy, and awkwardness were spotted in many places, such as “brainstormed ideas,” “One group of such limited studies has examined the effects of small-group student talk on L2 individual writing and reported inconsistent findings when comparing the planning with or without such talk….” Thus, the language and use of terms may be in need of vast revision or a professional proofreading service.

4. Section “Social Constructivism” may be moved to be a sub-section of “Literature Review.”

5. In the literature review, the author mentioned “Another group of studies has mainly examined such effects by comparing the use of small-group student talk in different discussion conditions, such as teacher-led vs. peer-led group talk….” Despite this paragraph being quite informative, it seems to me that this paragraph was somehow out of place, unconnected with the other parts of this paper, and unrelated to the research questions.

6. One key problem the study aims to address is the inconsistent findings of the previous studies on the effects of small-group student talk on L2 individual writing. However, to make the rationale of this study stronger, the author may need to specify how the present study could solve this problem. What may lead to the inconsistency? Why and how could one more empirical study be helpful in addressing the inconsistency if prior ones could not?

7. The sizes of the treatment group and control group are 24, while usually ANOVA is regarded as reliable only when the size of each group is bigger than 30.

8. The logic and coherence between and within paragraphs may be in need of major improvement, especially in the Discussion section.

Reviewer #2: 1. General comments. (1) Some sentences were too long to easily read, such as the second sentence in the abstract (To investigate …the comparison class); (2) some sentences were wordy (e.g., the first sentence in abstract and the opening sentence of the second paragraph in Introduction); (3) though there were not too many grammar errors, the language use was not such academic that should be refined carefully.

2. Comments to Introduction. The writing clue and logic were clear in this section in that the author(s) wrote in the general (talk) to specific manner (small-group student talk). Then, the author(s) introduced the existing relevant literature and identified the research gap. Some problems need to be concerned accordingly: (1) the literature described in the section did not have to be such long and explicit, you may describe key studies and point out the research potentials. The main contents of these studies could be put in Section 3; (2) in the end of this section, the author(s) noted the idea of ‘small-group student talk’ but offered no definition, which should be added. Because this is the key of this study.

3. Comments to Theoretical Framework. This section was written clearly. However, the author(s) could not only describe the theory per se, but also introducing how these theories were applied into the research. That is, the author(s) is/are recommended to explain how these theories could be used in the extant research (student talk and writing) and in this current study. The language in this section still needs refinement.

4. Comments on Literature Review. The author(s) comprehensively reviewed the studies of the effects of small-group student talk on L2 individual writing, indicating the understanding in this field. Whereas, some problems should be addressed: (1) the goal of this study is to investigate the sustained effects; thus, is it possible to summarize the test methods (e.g., posttest) used in the previous studies? (2) there was a lack of systematic comparison between the extant literature from the second to the third paragraph. [you may learn from this paper concerning how to systematically compare the extant literature. Su, F. & Zou, D. (2020). Technology-enhanced collaborative language learning: theoretical foundations, technologies, and implications. Computer Assisted Language Learning.] On the contrary, the literature reviewed seemed to be piled up without in-depth analysis; (3) I suggested the author(s) tabulate(s) a table to summarize the main review results, which could clearer present the research gap [you may learn from the following paper on how to present the main research findings of the most representative studies in the field. Zhang, R. & Zou, D. (2020). Types, purposes, and effectiveness of state-of-the-art technologies for second and foreign language learning. Computer Assisted Language Learning]; (4) language refinement is necessary.

5. Comments to Materials and Methods. (1) Information about the teacher of participants and the tasks of the teacher should be added in section 4.1.2 Participants; (2) how the teacher and the researchers collaborated in this experiment? [again, you may learn from this paper concerning how to describe the teacher-student collaboration. Su, F. & Zou, D. (2020). Technology-enhanced collaborative language learning: theoretical foundations, technologies, and implications. Computer Assisted Language Learning.] (3) the overlapped contents in Table 1 could be merged together for easy reading. Or the author(s) could transfer this table into a flow chart, which is more understandable; (4) to better orient readers, the authors(s) can add sub-headings for section 4.3, such as tasks in week one and tasks in week two, which could clearly present the experimental procedures and student tasks; (5) what were the teacher’s tasks in this experiment? (6) the authors(s) can tabulate a table to show the data analysis methods for the proposed two research questions. (7) why did not directly use the scoring rubrics for TEM-4, which is also authoritative; (8) how the two raters achieve agreements when discrepancies occurred in the scoring process. [you may learn from this paper concerning how to conduct assessment and grading. Zou, D. & Xie, H. R. (2018). Flipping an English writing class with technology-enhanced just-in-time teaching and peer instruction. Interactive Learning Environments]

6. Comments to Results and Discussion. This part was well written, but the language could be improved. (1) I wonder whether it is possible to present the results in the order of research question, such as the results of the first question and the results of the second question. [you may learn from this paper concerning how to present your research findings systematically. Chen, M.P., Wang, L.C., Zou, D., Lin, S.Y., Xie, H. & Tsai, C.C. (2019). Effects of captions and English proficiency on learning effectiveness, motivation and attitude in Augmented-Reality-enhanced theme-based contextualized EFL learning. Computer Assisted Language Learning.] (2) the author(s) made in-depth comparisons between the findings of this study and previous studies and offered reasonable explanation of the differences. The problem is that the author(s) did not use the afore-mentioned theoretical framework to analyze the research results; thus, more details should be given in this aspect.

7. The authors need to adjust the minor errors of the in-text citations. e,g, In the introduction (Line 1-4), “Talk is regarded as “the sea upon which all else floats” (Britton, 1970, p. 29) and has been considered as a key factor for all school learning, including writing, which is arguably the most challenging skill for learners in the process of learning a foreign language (Li, Zhang, & Parr, 2020; Zhang, 2013; Chen & Zhang, 2019).” It should be arranged according to alphabetical order.

8. It is suggested that the authors could arrange the part of literature review in a way that authors could easily understand rather than group all the literature into one section.

9. Current research focused on the effects of structured small-group talk on individual writing performance. The authors could elaborate more on the different effects of structured/naturally happened discussion instead of briefly introducing it in the data collection part.

10. The authors have discussed some formats of small-group talk as well as the effects of them on writing performance in the literature review. While this part seems irrelevant with the current research. Nothing related to the discussion format was mentioned in the latter part of the research. Which format was applied in the research? Did the structured small-group student talk was led by the teachers or the students? Why did we choose this format?

11. Since current research wanted to fill in the gap that there is limited research that has investigated the sustained effects of small-group discussion on writing performance, the authors could try to elaborate more on the sustained effects.

12. The question “Whether there is any difference in the effect of planning with small-group student talk and without it on students writing performance” has been raised. The authors need to briefly answer research question 2 in the conclusion.

6. PLOS authors have the option to publish the peer review history of their article (what does this mean?). If published, this will include your full peer review and any attached files.

Reviewer #1: No

Reviewer #2: No

---

## [Author Response · Author response to Decision Letter 0]

24 Mar 2021

Author Response to Reviewers

Dear editor and reviewers,

Thank you for your very constructive feedback, especially those critical comments that have really helped to streamline the focus of our paper. In our revised version, the focus is now brought to the fore, as expected. We have also successfully addressed all the other comments, queries, suggestions, which are now incorporated into the newly revised version. The large chucks of revised text are colour-coded, but the minor typographical changes are effected without being color-coded so that the reviewers can focus on reading the revised paper in its entirety. 

Sincerely in appreciation to your assistance

From authors

Reviewer #1: 

This study aims to investigate the effects of pre-writing small-group talk on L2 individual writing, which is interesting and highly relevant. The design of this study is structured and the instruments are reliable. To further improve this paper, the author may consider the points as follows:

1. Is the investigated topic, “small-group student talk before L2 individual writing,” equal to “brainstorming?” If so, the author may consider use the more concise expression; if not, the author may specify the difference between two.

Thank you very much for your professional and useful suggestions. We really appreciate all the time and effort you devoted to this paper at this special time. We sincerely hope you are staying healthy and safe. 

In this paper, “small-group student talk before L2 individual writing” is not equal to “brainstorming”. To our limited understanding of Osborn’s (1953) definition of brainstorming, it is a group or individual creativity technique by which attempts are made to gather ideas spontaneously contributed by its member(s) either in written or oral forms. Such a technique involves listing whichever ideas popping into mind about a specific topic without giving or receiving any criticism (Rakasiwi & Listyani, 2020). 

On the contrary, small-group student talk before L2 individual writing in this paper refers to the meaningful interactions among students in small groups who discuss L2 writing tasks prior to their individual writing, which may include content talk, language talk, organization talk, task-management talk, affective talk, and phatic talk (Li, Zhang, & Parr, 2020). Unlike the freely popped-out ideas that do not give or receive any criticism, such talk may give opinions on agreeing or disagreeing with the writing topic, generate and evaluate ideas and evidence that are for and against the writing topic, and select and organize ideas and evidence into a writing plan. 

Following your good advice, we have added an elaboration of the differences between them in the Introduction section on page 2. 

2. The author stated that “much less attention, so far, has been paid to small-group student talk before L2 individual writing, particularly when it comes to using structured small-group student talk to plan for L2 individual writing….” However, to my knowledge, there are many studies focusing on or at least involving the investigation of the effects of pre-writing planning/brainstorming.

Collectively, much is known about the use of student talk in pairs or small groups for developing students’ L2 writing skills in terms of when it occurs (during, or after writing) and what type of authorship it entails (collaborative or individual) (Storch, 2002, 2018; Storch and Wigglesworth, 2007; Watanabe and Swain, 2007; Shehadeh, 2011; Fernández Dobao, 2012, 2014; Storch and Aldosari, 2013; Yu, 2015; Xu, 2016; Yu and Lee, 2016; Xu and Kou, 2017, 2018). 

For studies targeting L2 pre-writing planning, the majority has laid emphasis on individual planning by investigating the effects of planning time, condition, language proficiency level, or task complexity on learners’ written texts (Ong and Zhang, 2010, 2013; Xing, 2015; Yi and Ni, 2015; Rahimi and Zhang, 2018, 2019; Wang and Zhang, 2019) rather than on examining leaners’ talk during their collaborative planning activities together with the effects of such talk to subsequent L2 individual writing. Comparatively, less is known about using small-group student talk as prewriting discussions for collaborative planning prior to L2 individual writing, particularly when it comes to structuring such talk and investigating the effects of such structured talk. 

We have taken your professional feedback and clarified this point in the summary paragraph of the Introduction part (please see p. 3-4 of the tracked-changes version).

3. The content may be preliminary good, but several parts of this article are somehow difficult to read. Redundancy, grammatical inaccuracy, and awkwardness were spotted in many places, such as “brainstormed ideas,” “One group of such limited studies has examined the effects of small-group student talk on L2 individual writing and reported inconsistent findings when comparing the planning with or without such talk….” Thus, the language and use of terms may be in need of vast revision or a professional proofreading service.

Thank you very much for pointing out the grammatical inaccuracy, redundancy, and awkwardness in our expressions. We have tried our best to carefully proofread every sentence of this paper. We hope it is better now. 

4. Section “Social Constructivism” may be moved to be a sub-section of “Literature Review.”

By following your good advice, we have moved the section of Social Constructivism to the Literature Review part (please see p. 4-5 of the tracked-changes version). 

5. In the literature review, the author mentioned “Another group of studies has mainly examined such effects by comparing the use of small-group student talk in different discussion conditions, such as teacher-led vs. peer-led group talk….” Despite this paragraph being quite informative, it seems to me that this paragraph was somehow out of place, unconnected with the other parts of this paper, and unrelated to the research questions.

After carefully reading and examining the connections between this paragraph with the rest of the paper, we realized that it deviated away from this paper. We then followed your good suggestion and deleted this paragraph. 

6. One key problem the study aims to address is the inconsistent findings of the previous studies on the effects of small-group student talk on L2 individual writing. However, to make the rationale of this study stronger, the author may need to specify how the present study could solve this problem. What may lead to the inconsistency? Why and how could one more empirical study be helpful in addressing the inconsistency if prior ones could not?

Thank you very much for bringing forward these problems. We have tried to clarify these issues in the Discussion section. Please see pages 18-19 of the tracked-changes version. We hope our explanations have addressed your concerns. 

7. The sizes of the treatment group and control group are 24, while usually ANOVA is regarded as reliable only when the size of each group is bigger than 30.

Considering the size number of each group was 24, we particularly examined normality, missing values and outliers prior to statistical analysis. We found no missing values and outliers. Besides, all the data of this study was normally distributed because the z-scores of skewness and kurtosis did not exceed 1.96 (Field, 2009) during the normality check. That’s why we decided to run ANOVA even the size number was smaller than 30. 

We have clarified this point in the section of Statistical analyses (please see p. 13 of the tracked-changes version).

8. The logic and coherence between and within paragraphs may be in need of major improvement, especially in the Discussion section.

We really appreciate your detailed reading and careful reviewing. We have revised the Discussion section accordingly (please see p. 17-20 of the tracked-changes version). 

Reviewer #2: 

1. General comments. (1) Some sentences were too long to easily read, such as the second sentence in the abstract (To investigate …the comparison class); (2) some sentences were wordy (e.g., the first sentence in abstract and the opening sentence of the second paragraph in Introduction); (3) though there were not too many grammar errors, the language use was not such academic that should be refined carefully.

Thank you very much for your professional feedback, especially the useful resources and good suggestions you shared with us. We really appreciate all the time and effort you devoted to this paper at this special time. We sincerely wish you safe, healthy, and happy every day. 

We have followed your useful advice and have revised the language and grammatical inaccuracy in the parts of Abstract and Introduction (please see p. 1-2 of the tracked-changes version). 

2. Comments to Introduction. The writing clue and logic were clear in this section in that the author(s) wrote in the general (talk) to specific manner (small-group student talk). Then, the author(s) introduced the existing relevant literature and identified the research gap. Some problems need to be concerned accordingly: (1) the literature described in the section did not have to be such long and explicit, you may describe key studies and point out the research potentials. The main contents of these studies could be put in Section 3; (2) in the end of this section, the author(s) noted the idea of ‘small-group student talk’ but offered no definition, which should be added. Because this is the key of this study.

We have taken your professional advice and have rewritten the Introduction part. Specifically, we added the definition of small-group student talk, maintained the key features of previous studies, clearly pointed out the research gaps, and highlighted the rationale of this study (please see p. 2-3 of the tracked-changes version).

3. Comments to Theoretical Framework. This section was written clearly. However, the author(s) could not only describe the theory per se, but also introducing how these theories were applied into the research. That is, the author(s) is/are recommended to explain how these theories could be used in the extant research (student talk and writing) and in this current study. The language in this section still needs refinement.

We have followed your good suggestions about the Literature part. In so doing, we have revised this part from three aspects, namely, introduction to the theoretical framework, relative research concerning student talk and writing within this framework, and the rationale for the current study (please see p. 4-5). 

4. Comments on Literature Review. The author(s) comprehensively reviewed the studies of the effects of small-group student talk on L2 individual writing, indicating the understanding in this field. Whereas, some problems should be addressed: (1) the goal of this study is to investigate the sustained effects; thus, is it possible to summarize the test methods (e.g., posttest) used in the previous studies? (2) there was a lack of systematic comparison between the extant literature from the second to the third paragraph. [you may learn from this paper concerning how to systematically compare the extant literature. Su, F. & Zou, D. (2020). Technology-enhanced collaborative language learning: theoretical foundations, technologies, and implications. Computer Assisted Language Learning.] On the contrary, the literature reviewed seemed to be piled up without in-depth analysis; (3) I suggested the author(s) tabulate(s) a table to summarize the main review results, which could clearer present the research gap [you may learn from the following paper on how to present the main research findings of the most representative studies in the field. Zhang, R. & Zou, D. (2020). Types, purposes, and effectiveness of state-of-the-art technologies for second and foreign language learning. Computer Assisted Language Learning]; (4) language refinement is necessary.

We really appreciate your detailed reading and careful reviewing. We have followed your advice and revised the Literature section accordingly (please see p. 3-7 of the tracked-changes version). 

5. Comments to Materials and Methods. (1) Information about the teacher of participants and the tasks of the teacher should be added in section 4.1.2 Participants; (2) how the teacher and the researchers collaborated in this experiment? [again, you may learn from this paper concerning how to describe the teacher-student collaboration. Su, F. & Zou, D. (2020). Technology-enhanced collaborative language learning: theoretical foundations, technologies, and implications. Computer Assisted Language Learning.] (3) the overlapped contents in Table 1 could be merged together for easy reading. Or the author(s) could transfer this table into a flow chart, which is more understandable; (4) to better orient readers, the authors(s) can add sub-headings for section 4.3, such as tasks in week one and tasks in week two, which could clearly present the experimental procedures and student tasks; (5) what were the teacher’s tasks in this experiment? (6) the authors(s) can tabulate a table to show the data analysis methods for the proposed two research questions. (7) why did not directly use the scoring rubrics for TEM-4, which is also authoritative; (8) how the two raters achieve agreements when discrepancies occurred in the scoring process. [you may learn from this paper concerning how to conduct assessment and grading. Zou, D. & Xie, H. R. (2018). Flipping an English writing class with technology-enhanced just-in-time teaching and peer instruction. Interactive Learning Environments]

Thank you very much for pointing out these problems. We have tried to clarify these issues in the Materials and Methods section. Please see pages 7-12 of the tracked-changes version. We hope our revisions have addressed your concerns. 

6. Comments to Results and Discussion. This part was well written, but the language could be improved. (1) I wonder whether it is possible to present the results in the order of research question, such as the results of the first question and the results of the second question. [you may learn from this paper concerning how to present your research findings systematically. Chen, M.P., Wang, L.C., Zou, D., Lin, S.Y., Xie, H. & Tsai, C.C. (2019). Effects of captions and English proficiency on learning effectiveness, motivation and attitude in Augmented-Reality-enhanced theme-based contextualized EFL learning. Computer Assisted Language Learning.] (2) the author(s) made in-depth comparisons between the findings of this study and previous studies and offered reasonable explanation of the differences. The problem is that the author(s) did not use the afore-mentioned theoretical framework to analyze the research results; thus, more details should be given in this aspect. 

After carefully reading and examining the issues you identified in our paper, we realized that we did not present our results in the order of research questions. However, considering that our study mainly highlighted the immediate and sustained effects of structured small-group student talk on L2 individual writing, we hope we can be allowed to keep the original way which first demonstrated immediate effects and then showed sustained effects. We also referred to all the research paper you suggested for reading and revised the Discussion section accordingly (please see p. 17-20 of the tracked-changes version). 

7. The authors need to adjust the minor errors of the in-text citations. e,g, In the introduction (Line 1-4), “Talk is regarded as “the sea upon which all else floats” (Britton, 1970, p. 29) and has been considered as a key factor for all school learning, including writing, which is arguably the most challenging skill for learners in the process of learning a foreign language (Li, Zhang, & Parr, 2020; Zhang, 2013; Chen & Zhang, 2019).” It should be arranged according to alphabetical order.

Thank you very much for your careful reading. We have followed your useful suggestion and revised the relevant parts (please see p. 2 of the tracked-changes version). 

8. It is suggested that the authors could arrange the part of literature review in a way that authors could easily understand rather than group all the literature into one section.

We have taken your professional feedback and rewritten the Literature Review section (please see p. 3-7 of the tracked-changes version).

9. Current research focused on the effects of structured small-group talk on individual writing performance. The authors could elaborate more on the different effects of structured/naturally happened discussion instead of briefly introducing it in the data collection part.

Thank you very much for your good advice. We have tried our best to add such elaboration in the Literature Review section. Please see pages 3-7 of the tracked-changes version. We hope our revisions have given you a better understanding. 

10. The authors have discussed some formats of small-group talk as well as the effects of them on writing performance in the literature review. While this part seems irrelevant with the current research. Nothing related to the discussion format was mentioned in the latter part of the research. Which format was applied in the research? Did the structured small-group student talk was led by the teachers or the students? Why did we choose this format?

After carefully reading and examining the issues you identified in this part, we realized that this paragraph was not closely related to the latter part of the research. We then followed your suggestion and deleted it (please see p. 3-7 of the tracked-changes version).

11. Since current research wanted to fill in the gap that there is limited research that has investigated the sustained effects of small-group discussion on writing performance, the authors could try to elaborate more on the sustained effects.

Thanks a lot for your professional feedback. We have added more elaboration on the sustained effects in the Discussion section (please see p. 20 of the tracked-changes version). 

12. The question “Whether there is any difference in the effect of planning with small-group student talk and without it on students writing performance” has been raised. The authors need to briefly answer research question 2 in the conclusion.

Thank you very much for reminding us to work on the coherence of the paper. We have followed your good suggestion and added our answers to the second research question both in the Discussion and the Conclusion sections (please see p. 17 and p. 21 of the tracked-changes version).

---

## [Decision Letter · Decision Letter 1]

29 Apr 2021

Effects of structured small-group student talk as collaborative prewriting discussions on Chinese university EFL students’ individual writing: A quasi-experimental study

PONE-D-20-31067R1

Dear Dr. Zhang,

We’re pleased to inform you that your manuscript has been judged scientifically suitable for publication and will be formally accepted for publication once it meets all outstanding technical requirements.

Kind regards,

Haoran Xie

Academic Editor

PLOS ONE

Additional Editor Comments (optional):

Reviewers' comments:

Reviewer's Responses to Questions

**Comments to the Author**

1. If the authors have adequately addressed your comments raised in a previous round of review and you feel that this manuscript is now acceptable for publication, you may indicate that here to bypass the “Comments to the Author” section, enter your conflict of interest statement in the “Confidential to Editor” section, and submit your "Accept" recommendation.

Reviewer #1: All comments have been addressed

Reviewer #2: (No Response)

2. Is the manuscript technically sound, and do the data support the conclusions?

Reviewer #1: Yes

Reviewer #2: (No Response)

3. Has the statistical analysis been performed appropriately and rigorously? 

Reviewer #1: Yes

Reviewer #2: (No Response)

4. Have the authors made all data underlying the findings in their manuscript fully available?

Reviewer #1: Yes

Reviewer #2: (No Response)

5. Is the manuscript presented in an intelligible fashion and written in standard English?

Reviewer #1: Yes

Reviewer #2: (No Response)

6. Review Comments to the Author

Reviewer #1: All my comments have been addressed. Considerable improvements have been spotted. I have no comment. I believe this paper will be valuable for educators and researchers who are interested in collaborative learning.

Reviewer #2: The author has made great efforts on the revised submission. The paper is ready for publication now.

7. PLOS authors have the option to publish the peer review history of their article (what does this mean?). If published, this will include your full peer review and any attached files.

Reviewer #1: No

Reviewer #2: No

---

## [Editor Report · Acceptance letter]

18 May 2021

PONE-D-20-31067R1 

Effects of Structured Small-Group Student Talk as Collaborative Prewriting Discussions on Chinese University EFL Students’ Individual Writing: A Quasi-Experimental Study 

Dear Dr. Zhang:

I'm pleased to inform you that your manuscript has been deemed suitable for publication in PLOS ONE. Congratulations! Your manuscript is now with our production department. 

Kind regards, 

on behalf of

Professor Haoran Xie 

Academic Editor

PLOS ONE